# Metastatic Adrenal PEComa: Case Report and Short Review of the Literature

**DOI:** 10.3390/medicina59010149

**Published:** 2023-01-11

**Authors:** Enrico Battistella, Luca Pomba, Marica Mirabella, Michele Gregianin, Antonio Scapinello, Marco Volante, Antonio Toniato

**Affiliations:** 1Endocrine Surgery Unit, Department of Surgery, Veneto Institute of Oncology, IOV-IRCCS, Via Gattamelata 64, 35128 Padua, Italy; 2Department of Nuclear Medicine, Veneto Institute of Oncology, IOV-IRCCS, Via Gattamelata 64, 35128 Padua, Italy; 3Pathology Unit, Veneto Institute of Oncology, IOV-IRCCS, Via Gattamelata 64, 35128 Padua, Italy; 4Department of Oncology, University of Turin, Via Giuseppe Verdi 8, 10124 Turin, Italy

**Keywords:** PEComa, adrenal incidentaloma, metastatic PEComa

## Abstract

PEComa has become a widely accepted entity, and increased recognition has led to descriptions of this tumor in a wide variety of anatomic sites, including the adrenal gland. PEComa (perivascular epithelioid cell tumor) is a mesenchymal tumor composed of perivascular cells, and the most frequent sites of PEComas are the uterus and retroperitoneum. The incidence is <1 per 1,000,000 people. We report a case of adrenal metastatic PEComa in a 63-year-old man discovered by a spontaneous hematoma of the rectus abdominis. In our case, PEComa of the adrenal gland was a significant diagnostic dilemma as the morphologic and immunophenotypic features of this neoplasm may easily be confused with those of other more commonly encountered lesions.

## 1. Introduction

PEComa of the adrenal gland may present a significant diagnostic dilemma as the morphologic and immunophenotypic features of this neoplasm may easily be confused with those of other more commonly encountered lesions. PEComa (perivascular epithelioid cell tumor) is a mesenchymal tumor composed of perivascular cells, and the most frequent sites of PEComas are the uterus and retroperitoneum. Since its first description, PEComa has become a widely accepted entity, and increased recognition has led to descriptions of this tumor in a wide variety of anatomic sites [1,2]. Among others, it can be found as an adrenal incidentaloma, an adrenal mass found during an imaging examination performed for other reasons, typically without clinical symptoms of adrenal disease. The etiology of adrenal incidentalomas includes benign and malignant lesions derived from the adrenal cortex, the medulla, or extra-adrenal tissues. The appropriate treatment for adrenal incidentaloma patients should be determined following discussion at a multidisciplinary expert team meeting [3,4,5]. Our case report discusses a very rare neoplasm that reserved notable diagnostic problems.

## 2. Case Report

A 63-year-old man with a spontaneous hematoma of the rectus abdominis muscles had an abdominal ultrasound (US), which revealed a 4 cm abdominal mass positioned in the left upper quadrant. Magnetic resonance imaging (MRI) to complete the evaluation of the lesion revealed a 4.1 cm round-shaped adrenal incidentaloma with intracellular fat and no indications of invasion. The patient’s functional activity was systematically assessed by means of endocrine biochemical testing, regardless of the presence or absence of symptoms. Measurement of urinary catecholamines and metanephrines, 24 h urine-free cortisol (UFC) excretion, late-night salivary cortisol (LNSC) levels, and 9 am plasma cortisol following an overnight dexamethasone suppression test resulted negative. Hypokalemia was not demonstrated. Following the examination, the patient was referred to our outpatient endocrine surgery clinic. In anamnesis, the patient presented the following comorbidities: obesity (Body Mass Index = 38), myasthenia gravis, atrial fibrillation with anticoagulant therapy, and previous cardiac surgery for pulmonary valve stenosis. He denied having abdominal pain, and he did not complain about nausea, vomiting, fever, chills, anorexia, or weight loss. Abdominal examination was normal, with no tenderness or evidence of hepatosplenomegaly. The patient also had a DOPA PET scan and an FDG PET scan (fluorodeoxyglucose positron emission tomography), which were both negative.

The case was discussed in a multidisciplinary meeting, and a conservative approach was recommended for important comorbidities, including an enhanced MRI (Magnetic Resonance Imaging) of the abdomen at 12 months and biochemical testing. Following a new MRI one year later, the lesion appeared to have grown in size, measuring 5.9 cm in diameter without evidence of malignancy or infiltration. Based on the increase in tumor size, the case was re-discussed, and left laparoscopic adrenalectomy was undertaken with the patient’s consent. The surgical intervention started with a transabdominal lateral laparoscopic approach but, since its beginning, it was complicated with frequent minor hemorrhages from the lesion, necessitating conversion to a subcostal laparotomy. The mass was excised en bloc with the entire capsule. The patient was discharged without complications on POD #5. Histological diagnosis was of adrenocortical adenoma (Weiss Score: 1 point because of a nuclear grade of 3 points according to Fuhrman criteria). No vascular or peritumoral capsular invasions were observed. The mitotic index was <1 × 10 HPF.

The patient underwent an annual follow-up for 3 years with imaging (computed tomography—CT or MRI), which was negative for any pathologic features other than muscular relaxation at the site of the laparotomic incision.

Three years later, the patient presented to the emergency unit with walk instability and lower back pain that had persisted for fifteen days without any relief despite medications. Neurological examination revealed symmetrical reflexes in the biceps, triceps, and a left prevalence in the knees and ankles. Upright posture could be maintained for a few seconds with double support. The patient had a lumbar spine MRI, which revealed an expansive lesion with medullary compression in D9, as well as a smaller lesion in D11. Emergency neurosurgery was performed on the patient, and a decompressive laminectomy of the D8–D11 vertebra was carried out. The histological report concluded that the patient had epithelioid mesenchymal neoplasia with clear and eosinophilic cells consistent with bone metastases from PEComa. Pathological review of the adrenal mass resulted in a revised diagnosis of adrenal PEComa, with an immunophenotype melan A +, cathepsin K +, synaptophysin −, chromogranin A −, calretinin −, SF1 −, HMB45 −, smooth muscle actin −, and inhibin − (Figure 1 and Figure 2).

FISH test for the detection of the TFE3 (Xp11.23) rearrangement was negative in both adrenal and bone lesions.

A new FDG-PET revealed a round-shaped enhanced mass in the left para-aortic region compatible with lymph node enlargement, multiple areas of elevated SUV in the lumbar spine, and a single 8 mm nodule in the right upper pulmonary lobe (Figure 3).

The patient died of cardiac failure one month later.

## 3. Discussion

Adrenal incidentaloma is defined as a clinically inapparent adrenal mass (>1 cm in diameter) found during an imaging study performed for a reason unrelated to adrenal disease. This definition excludes patients who are undergoing screening and surveillance because of hereditary syndromes or those with known extra-adrenal cancer who are undergoing imaging for staging or during follow-up after treatment. The prevalence of adrenal incidentaloma has been reported to be 1 to 6% and is higher among older adults (V–VII decades), regardless of gender. According to the literature, 80% of incidentalomas are non-functioning adenomas, 5% are subclinical Cushing’s syndrome, 5% are pheochromocytoma, 2.5% are adrenal metastases, and the remaining are adrenal carcinomas, PGL, myelolipomas, or benign cysts [3,4,5]. Patients diagnosed with adrenal incidentaloma should undergo a thorough evaluation that includes clinical examinations, laboratory tests, and imaging studies to ascertain the nature of the adrenal mass.

According to Mantero et al., a non-functional adrenal incidentaloma that is 4 cm or less and has been positively identified in imaging tests should be checked periodically via radiological imaging (usually benign lesions present a low CT-attenuation, <10 HU) and hormone evaluation. Surgical treatment should be performed regardless of tumor size, radiological imaging features, and in all cases of hormone-hypersecreting tumor-associated clinical symptoms [3,4,5,6,7]. Furthermore, a non-functioning adenoma larger than 4 cm and with an increase in size observed during a follow-up period of 4 years should be surgically removed due to the possibility of malignancy [7,8]. In our case report, the adrenal mass detected in the first MRI was found to be borderline in size for upfront surgical treatment, and given the patient’s comorbidity, a multidisciplinary team recommended conservative treatment. After an increase in size was observed in the follow-up, the case was re-discussed with a recommendation for adrenalectomy. The patient underwent complete surgical resection, and the histological report indicated adrenal adenoma. It was only after the patient underwent emergency surgery, and the histological specimen revealed a metastasized PEComa, that a review of the adrenal specimen supported the non-cortical origin of the lesion and was coherent with a primary adrenal PEComa. Globally, there is ≤1 malignant PEComa diagnosis per 1,000,000 people annually, with an estimated 100–300 new patients per year in the United States [1,2]. The most frequent sites of PEComas are the uterus and retroperitoneum, though the tumor can be found virtually anywhere [9]. The present case represents one of few examples of malignant PEComa that arise in the adrenal gland. In the literature, only three cases have been reported in the adrenal gland to the best of our knowledge [10,11]. The literature reports that PEComa has a higher incidence in females (6–7:1), and the age at diagnosis ranges from 8–89 years. PEComa (perivascular epithelioid cell tumor) was coined by Bonetti in 1992 to describe mesenchymal tumors composed of perivascular cells that exhibit focal association with blood vessel walls and typically express melanocytic and smooth muscle markers such as HMB-45, HMSA-1, MelanA/mart1, microphthalmia transcription factor (MITF), actin, and desmin [11,12]. Folpe described high-risk criteria for malignancy as having a size > 5 cm, a mitotic rate > 1/50 hpf, necrosis, a high nuclear grade and cellularity, vascular invasion, and an infiltrative growth pattern. If two or more high-risk features are present, it is classified as malignant [13]. A PEComa of the adrenal gland can be extremely difficult to differentiate from an adrenocortical neoplasm, metastatic carcinoma from an extra-adrenal site, a smooth muscle tumor, and a clear cell sarcoma, which can present with substantially similar histology. Immunohistochemical findings might be worrisome in this setting of differential diagnosis due to some possible immunophenotypic overlap (i.e., melanA expression in both PEComa and adrenocortical tumors). PEComas tend to recur locally or to develop distal metastases, most commonly in the lung. Metastatic spread can occur even 7 years after curative surgery [14,15]. The gold standard treatment for PEComas is primary surgical excision with the aim of achieving negative margins. Complete resection is critical for determining tumor histopathological risk factors. Several protocols of systemic chemotherapy have been used with little efficacy. A dearth of reported cases makes it difficult to determine the true impact of PEComas and the optimal chemotherapy schedule. Surprisingly, good survival rates without therapy were also reported for up to one year after diagnosis. Furthermore, the role of radiation therapy remains unclear [16,17,18].

## 4. Conclusions

Adrenal PEComa is a very rare pathology with a varying prognosis based on histological features. The variety of therapeutic strategies used with heterogeneous results, as well as the lack of established guidelines, highlights the need for randomized trials. While the optimal course of treatment is debatable, surgical resection remains the cornerstone of therapy.

## Figures and Tables

**Figure 1 medicina-59-00149-f001:**
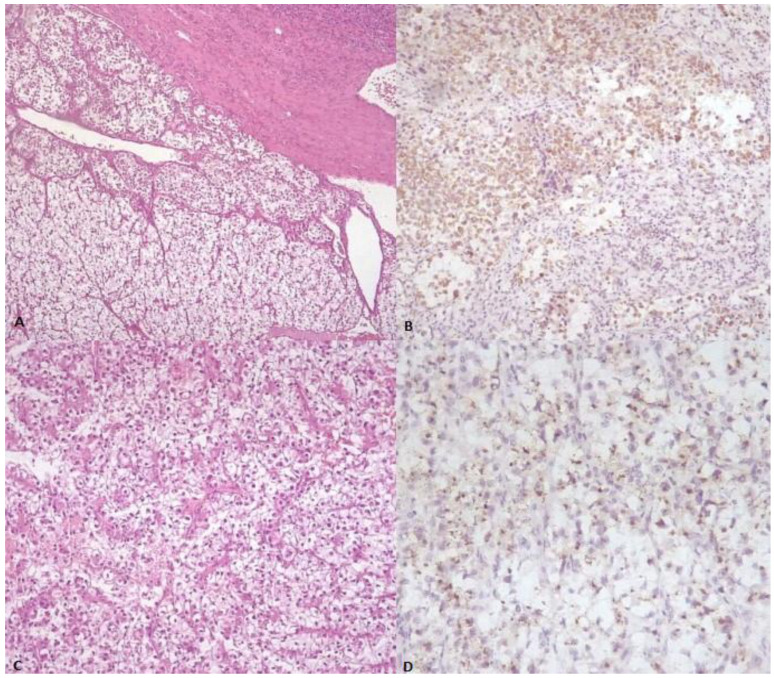
(**A**) Low-power image demonstrating sheets of neoplastic cells with abundant clear cytoplasm (center), almost indistinguishable from normal adrenal gland tissue (bottom) (H&E stain). (**B**) The neoplastic cells stained positively with antibody against Melan-A. (**C**) Population of neoplastic cells showing round nuclei and abundant clear cell and eosinophilic cytoplasm, partially surrounded by thin and delicate, curved vessels (H&E stain). (**D**) The tumor cells exhibit focal granular cytoplasmic immunoreactivity for cathepsin K.

**Figure 2 medicina-59-00149-f002:**
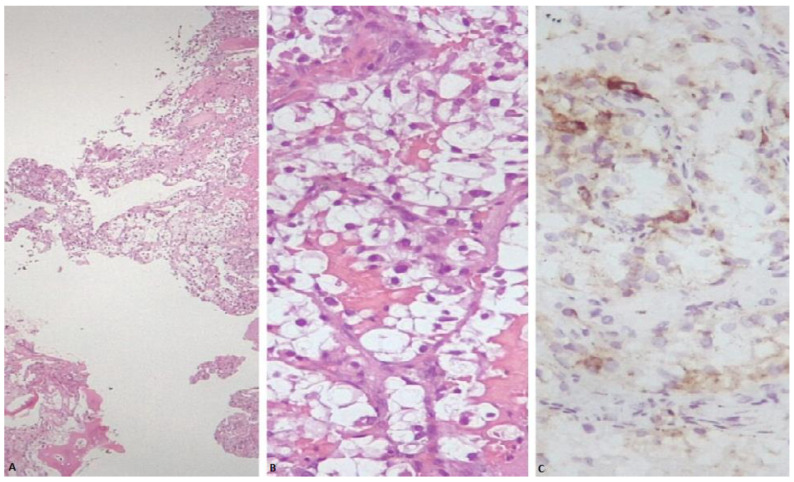
(**A**) Sheets of neoplastic cells showing abundant clear cytoplasm, infiltrating plates of trabecular bone (H&E stain). (**B**) Nests of neoplastic cells showing round to oval nuclei and abundant clear cell cytoplasm, partially surrounded by a thin and delicate vasculature (H&E stain). (**C**) Neoplastic cells within bone exhibiting focal granular cytoplasmic immunoreactivity for cathepsin K.

**Figure 3 medicina-59-00149-f003:**
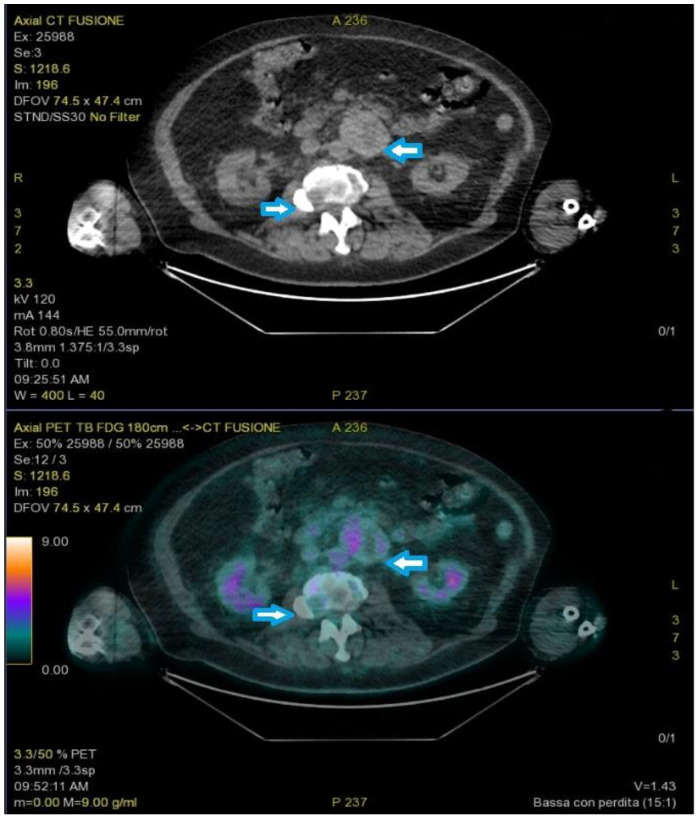
PET-FDG showed a round-shaped enhanced mass in the left para-aortic region and multiple areas of elevated SUV in the lumbar spine (arrows).

## Data Availability

Data was obtained from the patient’s medical records.

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
