# Peer review of "Metastatic Adrenal PEComa: Case Report and Short Review of the Literature"

_medicina, 2023, doi:10.3390/medicina59010149_

Round 1

Reviewer 1 Report

Dear Authors, 

This article is interesting because it deals with a rare tumor in an even rarer location and highlights the difficulty of anatomopathological diagnosis.

To my knowledge, you have cited described cases of PEComa of the adrenal gland

I have no major comment, apart from reference problems.

Line 66 : point : write twice

Line 135 : the reference 6 does not correspond

Line 136 : missing reference 9 for adrenal PEComa

Line 153 : missing reference for metastatic PEComa

I do not understand the interest of reference 17 which is not cited in the text

Author Response

Answer to reviewer: Reference problems.

1- Line 66 : point : write twice

It was removed

2- Line 135 : the reference 6 does not correspond

Line 136 : missing reference 9 for adrenal PEComa

Line 153 : missing reference for metastatic PEComa

I do not understand the interest of reference 17 which is not cited in the text

  • There was a mistake in the sequence of number in the references. We provide to rectify the references section

Reviewer 2 Report

The paper entitled Metastatic adrenal PEComa: Case report and Short Review of the Literature is a typical case report description. The value of this publication is related to the high rarity of Pecoma in the adrenal gland.

I have two comments about the publication.

The first is the fact that the first descriptions related to Pecoma were given 4 years earlier than in the cited publication under Zamboni's surrname. Although in 1992 Bonetti  (Bonetti, F.; Pea, M.; Martignoni, G.; Zamboni, G. PEC and sugar. Am. J. Surg. Pathol. 1992, 16, 307–308) was the first to describe this type of tumor, it was Zamboni (Zamboni, Giuseppe, et al. Clear cell “sugar” tumor of the pancreas: a novel member of the family of lesions characterized by 194 the presence of perivascular epithelioid cells. Am. J. Surg. Pathol., 1996, 20.6: 722-730) who actually introduced the name Pecoma to the literature.

The second point is broader. The authors focused solely on the rarity of Pecoma in the adrenal gland in relation to other incidentalomas in the adrenal gland. In my opinion, for a fuller value of the publication, other Pecoma locations should be mentioned. Pecomas are found more commonly in the uterus.

Author Response

Answers to reviewer:

1- The first is the fact that the first descriptions related to Pecoma were given 4 years earlier than in the cited publication under Zamboni's surrname. Although in 1992 Bonetti  (Bonetti, F.; Pea, M.; Martignoni, G.; Zamboni, G. PEC and sugar. Am. J. Surg. Pathol. 1992, 16, 307–308) was the first to describe this type of tumor, it was Zamboni (Zamboni, Giuseppe, et al. Clear cell “sugar” tumor of the pancreas: a novel member of the family of lesions characterized by 194 the presence of perivascular epithelioid cells. Am. J. Surg. Pathol., 1996, 20.6: 722-730) who actually introduced the name Pecoma to the literature.

  • We provide to add in the references the paper and to rectify in the manuscript  as suggested.

2- The second point is broader. The authors focused solely on the rarity of Pecoma in the adrenal gland in relation to other incidentalomas in the adrenal gland. In my opinion, for a fuller value of the publication, other Pecoma locations should be mentioned. Pecomas are found more commonly in the uterus

  • Line 135-136: "The most frequent sites of PEComas are the uterus and retroperitoneum, though the tumor can be found virtually anywhere "